# High-Precision and Low-Cost Wireless 16-Channel Measurement System for Malachite Green Detection

**DOI:** 10.3390/mi9120646

**Published:** 2018-12-07

**Authors:** Tong Shen, Tong Zhou, Ying Wan, Yan Su

**Affiliations:** School of Mechanical Engineering, Nanjing University of Science and Technology, Nanjing 210094, China; shentong@njust.edu.cn (T.S.); yingwan@njust.edu.cn (Y.W.); yansu@njust.edu.cn (Y.S.)

**Keywords:** DNA biosensor, electrochemical instrument, malachite green detection

## Abstract

Focusing on the issue of the malachite green traditional test methods such as large volume, high cost and high complex, this paper proposed a novel multi-channel electrochemical malachite green detection system. Specific recognition properties of malachite green DNA adapter is employed to realize accurate sensing of concentration of malachite green, which can achieve precise detection of malachite green concentration with low noise and high precision. The maximum measurement capability of multi-channel acquisition system is 16 samples in a batch. According to the experimental results, malachite green could be detected quantitatively in the range from 10^−3^ μg/mL to 10 μg/mL, which performs well in the test of malachite green residues in aquatic product transportation.

## 1. Introduction

Malachite green (MG) is extensively adopted in aquatic transportation and large-scale aquaculture, because of its excellent sterilization characteristics on the water mildew of the fish and fish eggs [1]. However, MG is injurious to human health through the food chain [2] with the effects of carcinogenicity and teratogenicity. In consideration of its harmfulness, the European Union has restricted the residual concentration of MG less than 2 mg/L. As a kind of antibiotics, MG is also banned by the department of agriculture.

There are several methods to detect MG, such as high-performance liquid chromatography [3], enzyme-linked immunoassay [4], ultraviolet spectrophotometry [5] and so forth. Although these methods have been developed for many years, there are several drawbacks including high assay cost and long assay time, which restrict their application. Therefore, it is of great significance to construct a fast, sensitive and convenient residual detection system for MG.

In recent years, electrochemical detection method [6,7] has played an important role in antibiotics detection, which enjoys the advantages of high accuracy, strong analytical ability and fast speed. A complementary metal-oxide-semiconductor (CMOS) single-chip electrochemical array system has been proposed in Reference [8], which is suitable for the implementation of a fully integrated instrument-on-chip microsystem. R.F.B. Turner has proposed a simple CMOS integrated potentiostatic control circuit [9], which converted the chemical concentration into a proportional voltage. However, the drawbacks of high cost, less precision and less user-friendly remains unaddressed effectively [10,11,12]. Moreover, these electrochemical instruments’ unsystematic and wired transmission mode cause the inconvenience during the motion detection. Therefore, it is of great significance to design a kind of rapid and portable instrument for malachite green detection.

Since the seminal work by Ruojun Lao et al. introducing high-performance electrochemical DNA-based sensors, significant progress in the development of DNA-based sensors has been achieved [13,14,15,16]. DNA has a strong specific recognition ability and is more stable and non-degradable than RNA and it is not required to use diethyl pyrocarbonate (DEPC) treatment like RNA sensor [17]. Moreover, the sensor constructed in this work is not only accurate but also easy to operate. Furthermore, a 16-channel electrochemical instrument with features of high-precision, low-cost and user-friendly for MG detection is also developed. Two electrochemical methods of cyclic voltammetry and amperometry are integrated in this instrument, which could measure the electrochemical currents in the range of microamperes to nanoamperes. The measurement system could assess the quality of the samples by testing on the spot and send the test results wirelessly to the cloud server by the general packet radio service (GPRS) [18], which possesses well portability.

## 2. Method and Architecture of the Measurement System

A novel measurement system is established to detect the concentration of MG in aquatic transportation, as shown in Figure 1. The system is composed of 16 DNA biosensors and a hand-held electrochemical instrument with a GPRS module. The biosensors are driven by the hand-held electrochemical instrument to generate the corresponding inductive currents, which are equipped with the three-electrode system [19]. The design of three-electrode system is similar to previously reported [20], it contains a working electrode (WE), an auxiliary electrode (AE) and a reference electrode (RE). WE is an electrochemical reaction place and generates polarization current. AE provides the current loop for the WE. RE functions as a potential reference with no actual electrochemical reaction. The hand-held electrochemical instrument is exploited to collect the response current of the 16 sensors and calculate the concentration of the measured substance. Finally, GPRS module is applied to send the data to the cloud server. Users can access the cloud server and check the detection information through the smart phone.

### 2.1. Construction of the DNA Biosensor

The materials involved in DNA biosensor are as follows: MG and gold (III) chloride trihydrate (HAuCl_4_), which are obtained from Sigma Aldrich (St. Louis, MO, USA). DNA oligonucleotides are synthesized from Shanghai Sangon Biotech Co. Ltd. (Shanghai, China). The sequence of the thiolated MG aptamer1 (thiolated Apt1) is 5′-SH-(CH_2_)_6_-TGG AAG GAG GCG TTA TGA GGG GGT CCA CG-3′ (thiol and a -(CH_2_)_6_- alkyl chain is modified at its 5-terminal) and the MG aptamer2 (bio-Apt2) is 3′-biotin-GCA CCT TCC TCC GCA ATA CTC CCC CAG GTG C-5′ (biotin is modified at its 3-terminal). The target can perfectly match to the sequence of the probe. Avidin-horseradish peroxidase (Av-HRP) is purchased from Roche Diagnostics (Mannheim, Germany). The buffer solution for the measurement is TMB substrate (Neogen K-blue low-activity substrate). The washing buffer is PBS buffer (0.1 M, pH 7.4). Av-HRP is diluted by using 0.1 M PBS buffer including 0.5% casein (pH 7.4). All solutions are prepared with ultrapure water (electric resistivity >18 MΩ) from a Millipore Milli-Q water system.

The electrochemical DNA sensor is constructed on the screen printing electrode. Screen printing is a valuable technique that allows the design of electrodes with an enhanced surface area that can be further functionalized with organic and bioorganic receptors. Screen printed materials have recently found applications in dye-sensitized solar cells [21], electrochromic devices [22] and sensors [23]. Our screen printed electrodes combine small volume and low cost. The construction process of the biosensor is shown in Figure 2. Firstly, the screen printing electrode (SPE) is cleaned with ultrapure water for 1 min and dried subsequently. Then, the 1 mg/mL HAuCl_4_ solution is restored by amperometry with the scanning voltage of −0.2 V and the scanning time of 100 s to generate AuNPs on the surface of the WE, so that the thiolated Apt1 could be immobilized more stable on the electrode surface. The electrodes are rinsed three times with PBS buffer (pH 7.4) and dried after the scanning.

Then 3 μL 1 μM thiolated Apt1 is fixed on the WE at 4 °C. After fixation, the electrode is rinsed with PBS buffer (pH 7.4) for 1 min and dried subsequently.

After the immobilization of the aptamer1, the mixed solution of 3 μL malachite green and 3 μL bio-Apt2 at 1 μM is uniformly fixed on WE surface. After 1 h incubation at 37 °C, a “sandwich” structure is formed with Apt1, target (MG) and Apt2.

The WE is treated with Av-HPR after rinsed and dried. Electrode is then rinsed with PBS buffer (pH 7.4) and subjected to electrochemical measurements after 20-min incubation at room temperature. Av-HRP is attached to the biotin labels via the biotin-avidin bridge.

Finally, cyclic voltammetry (CV) and amperometry are employed to characterize the enzyme based on electrocatalytic process in the TMB solution with H_2_O_2_.

### 2.2. Hand-Held Electrochemical Instrument

The hand-held electrochemical instrument possesses three functions: providing the necessary scanning voltage between the AE and the RE to control the voltage between WE and RE; collecting the induction current signals on the WE and converting them into the corresponding concentration value; transmitting the test results to the cloud server via GPRS module.

Taking one channel’s detection process as an example, the measurement system structure schematic is shown in Figure 3. The digital-analog converter (DAC) is controlled by microcontroller unit (MCU, STM32) to generate the specified scanning voltage, which is applied between the biosensor’s WE and RE by the constant potential circuit. The current detection circuit converts the response current on the WE into the voltage. The analog-digital converter (ADC) samples the voltage and transmits them to the MCU. After data processing, MCU send the test results via GPRS module.

Two electrochemical methods are integrated in the hand-held electrochemistry instrument: CV and amperometry. CV is applied to qualitatively analyze the samples by the reducing peak and amperometry is applied to quantitatively analyze samples. CV is performed on the three-electrode system by the electrochemical instrument firstly. The initial potential is 0 V, the maximum potential is 0.7 V, the final potential is 0 V, the scanning rate is 0.1 V/s and the scanning period is one cycle. Amperometry is performed subsequently at 0.1 V scanning potential and 100 s scanning time. The response current is sampled by the instrument at the end of amperometry. Finally, the corresponding concentration value is calculated by MCU according to the current-concentration equation.

## 3. Characteristics of Measurement System

### 3.1. High Detection Accuracy

The electrochemical detection principle of this system is that TMB is oxidized by H_2_O_2_ to azo compounds under the catalyzed by HRP, azo compounds and H_2_O is generated simultaneously. Hence, in this three-electrode system, as the electrochemical reaction going on, the concentration of reactants (HRP, H_2_O_2_ and TMB substrate) on the electrode surface is decreasing and the concentration of products (azo compounds) on the electrode surface is increasing, which result in deviation from the set potential between the WE and the RE. Therefore, the amplifier is applied to ensure the voltage stability between WE and RE, which is dynamic adjusted to avoid change with the electrochemical reaction. However, the RE will be polarized once current flowing through and the potential on the RE will be changed with the flowing current. In order to ensure the stability of the potential between WE and RE, operational amplifier is adopted before the RE to generate a high input impedance range from 10^12^ Ω to 10^15^ Ω. In this case, the current flowing through the RE can be ignored, which will greatly improve the stability of potential between RE and WE. The circuit for the control amplifier is designed as shown in Figure 4.

In the circuit, U1 functions as a voltage follower to obtain high input impedance and low output impedance, which possesses excellent isolating and buffering effect on the front and rear levels. U3 is placed in front of the RE to greatly increase the input impedance and avoid the influence of the subsequent amplifier circuit as well as a voltage follower. U2 serves as a reverse proportional amplifier, forming a negative feedback circuit with U3 and the equivalent impedance Z2. The equivalent impedance of the three-electrode system is also shown in Figure 4. Z1 is the equivalent impedance between the RE and the WE and Z2 is the equivalent impedance between the RE and the AE. Capacitance C1 effectively prevents the self-excited oscillation of operational amplifier. AD8609 is a wide bandwidth, precise operational amplifier, with typical 40 μV offset voltage and maximum 4.5 V/°C imbalance drift.

The induced current signal (ionic current) from the WE is converted to voltage by U4. The magnitude of the induced current signal on the electrode depends on the electrical activity of the electroactive analyte in the reaction substrate. Due to the high input impedance, the current flowing through RE is almost zero. Therefore, the induced current is approximately equal to the current flowing through the WE. The output voltage is $V_{o}=I_{WE}\cdot R_{f}+2.5$. Feedback resistance *R_f_* is a low-noise, high-precision and low-temperature coefficient metal film resistor, which is applied to improve the accuracy and sensitivity of current detection.

Compared with the basic potentiostat circuit which consists of two operational amplifiers (U2 and U4), the potentiostat in this work is more accurate. For the standard potentiostat circuit, when the equivalent resistance Z1 changed, the potential of the RE will change correspondingly, thus affecting the potential between WE and RE. And there’s a current flowing through the RE, which reduced the detection accuracy greatly. Compared with commercial potentiostat system and microchips, such as LMP9100, the potentiostat designed in this work has higher precision and low noise. Due to the low MG detection limit, the sensitivity and precision need to be higher. This requires the voltage between the working electrode and the reference electrode of the sensor more stable and the reading of the weak induction current produced by the working electrode more accurate. Hence, the potentiostat in this work has more advantages for MG detection.

However, in the practical implementation, there is a parasitic capacitance C_S_ between the input terminals, which is comprised by the input capacitance of the operational amplifier and the distributed capacitance of the wiring, as shown in Figure 4. Under the influence of the first-order lag network which is constituted with the input capacitance C_S_ and the feedback resistance $R_{f}$, it is easy to generate the self-excited oscillation on the current amplifier, thereby restricting its frequency characteristic. As shown in Figure 5a, in the condition of without lead correction, the intersection point between open-loop gain and noise gain is $\left|A(f)\right|=\left|G_{N}(f)\right|$, when |AF|=1, the intersection of open loop gain and the noise gain is considered as the critical frequency $f_{AF}$. When the signal frequency exceeds $f_{AF}$, the self-excited oscillation may occur. Therefore, the feedback capacitance $C_{f}$ is utilized to compensate the distributed capacitors existing in the circuit, thereby avoiding the self-excited oscillation and improving the stability.

The noise gain of the amplifier with lead correction can be calculated:(1)$$G_{N}\left(f\right)=1+\frac{R_{f}\|\frac{1}{2\pi fC_{f}}}{\frac{1}{2\pi fC_{S}}}=1+\frac{2\pi R_{f}C_{S}f}{1+2\pi R_{f}C_{f}f}=\frac{1+2\pi R_{f}\left(C_{f}+C_{S}\right)f}{1+2\pi R_{f}C_{f}f}=\frac{1+\frac{f}{f_{z}}}{1+\frac{f}{f_{p}}}$$

In the formula, $f_{z}$ and $f_{p}$ represent the zero frequency and the pole frequency of noise gain (*G_N_*) respectively and $f_{z}<f_{p}$. In order to avoid the self-excited oscillation of the amplifier in the low frequency bandwidth, the pole frequency $f_{p}$ should be less than the critical frequency $f_{AF}$, as shown in Figure 5b.

Under the condition of f much larger than the $f_{p}$, the noise gain can be simplified:(2)$$G_{N}\left(f\gg f_{p}\right)=\frac{C_{f}+C_{S}}{C_{f}}$$

Due to the relationship among the gain bandwidth (*GBW*), the open loop gain and the frequency of operational amplifier is: GBW=f⋅A(f), the feedback capacitance can be calculated:(3)$$\begin{array}{c} \left|A\left(f_{AF}\right)\right|=\frac{GBW}{f_{AF}}\\ \left|G_{N}\left(f_{AF}\right)\right|=\frac{C_{f}+C_{S}}{C_{f}}\\ \left|A\left(f_{AF}\right)\right|\geq \left|G_{N}\left(f_{AF}\right)\right|\\ f_{AF}=\frac{1}{2\pi R_{f}C_{f}} \end{array}\}\Rightarrow C_{f}\geq \frac{1+\sqrt{1+8\pi R_{f}C_{S}\cdot GBW}}{4\pi R_{f}\cdot GBW}\approx \sqrt{\frac{C_{S}}{2\pi R_{f}\cdot GBW}}$$

Hence, $\left|A\left(f\right)\right|\geq \left|G_{N}\left(f\right)\right|$ can be valid at the condition Cf≥CS2πRf⋅GBW, thereby preventing the self-excited oscillation of the mutual resistance amplifier. Here, $C_{f}$ is set to 47 pf.

### 3.2. Multichannel Measurement Method

Multi-channel detection method [24] is employed in the system. Considering the factors of signal to noise ratio (SNR), linearity and channel number, ADS1258 is selected as the ADC, which enjoys advantages of high-precision and low-power. The multi-channel measurement circuit diagram is shown in Figure 6. AD8629 is served as voltage follower to increase power of the output voltage and match the impedance of the back-end circuits. ADC is controlled by STM32 to sample the signals from 16 modules and the 2.5 V reference voltage is provided by the reference source ADR421. In order to improve the measurement accuracy and reduce the circuit noise, a third-order butterworth low-pass filter is exploited to filter out signals that are more than 30 Hz to avoid the leakage of high frequency signals.

## 4. Results and Discussion

### 4.1. Multichannel Precision Test

The adjustable precision current regulator with extremely low noise was applied to evaluate the weak current detection capability of the system. The current regulator’s output ports were connected with the working electrode interfaces and the common ports were connected with the interfaces of RE and AE. After that, 12 groups of constant current values were respectively set as 2 mA, 0.7 mA, 700 μA, 200 μA, 70 μA, 20 μA, 7 μA, 2 μA, 200 nA, 80 nA, 12 nA and 1 nA to test. The test results of 16-channel constant current detection are shown in Table 1. In this table, channel fluctuation is the ratio of standard deviation to average current of channels.

The test results indicate that proposed method is with high precision and channel consistency. When measured values are over 100 nA, the maximum current measurement error is <1% and the fluctuation among channels is <0.34%. When measured values are in a range from 10 nA to 100 nA, the maximum current measurement error is <3.5% and the fluctuation among channels is <1%. When measured values are <10 nA, the maximum current measurement error is <10% and the fluctuation among channels is <10%.

### 4.2. Biosensor Test

The biosensors were tested in different concentration of target by CV and amperometry to validate the performance of the biosensor. Firstly, CV was employed to test a 1 mg/L malachite green sample and a 0 mg/L sample respectively, by commercial electrochemical Instrument purchased from CH Instruments Inc. As shown in Figure 7a, when the scanning voltage is around 0.25 V, the reduction peak has been increased significantly after the recognition of target, which indicated that Av-HRP has been successfully attached to the working electrode surface, which also indicated the existence of MG in the sample. For comparison, the same sample were tested by the designed instrument as well, as shown in Figure 7b. It is indicated that the experiment data agrees very well with the data detected by the commercial electrochemical instrument in CV. In order to quantitatively measure the target, MG solution with different concentrations (1 mg/L, 100 μg/L, 10 μg/L, 1 μg/L and 0 μg/L) were detected by amperometry. For comparison, the same solutions were tested by both the commercial electrochemical instrument and the designed instrument respectively, as shown in Figure 7c,d, the current values in a period of 100 s were obtained by the instrument. It is indicated that the experiment data agrees very well with the data detected by the commercial electrochemical instrument in amperometry. We found that the amperometric signal was logarithmically related to the sample concentration in a range from 1 μg/L to 1 mg/L, which spanning a response region of at least 3 orders of magnitude, as shown in Figure 7d. The regression equation of y=2.736x+2.338 (R^2^ = 0.992) was obtained, where *y* is the amperometric signal value in μA and *x* is the logarithmic concentration of target in μg/L.

In order to further determine the performance of the sensor in the actual sample detection, the reliability, reproducibility and stability of the biosensor were carried out and were compared with the RNA aptamer-based electrochemical biosensor in Reference [17]. The sensor designed in this system was adopted to detect the mixture of 10 mg/mL MG and 100 mg/mL other structural analogs (including chloramphenicol, cryptochrome malachite green, eugenol, etc.). The current variation between the mixed solutions with the interfering substances and those without interfering substances was less than 3.7% but 4.0% in Reference [17], which confirmed the well reliability of the fabricated biosensor in this system. In order to estimate the reproducibility, five sensors with the same assembly step electrodes were used to detect MG with 0.1 mg/mL concentration. The test results indicated the relative standard deviation (RSD) is 3.3%, compared to 3.4%in Reference [17]. Moreover, we utilized five electrodes fabricated under the identical conditions and placed them in the 4°C fridge for 2 weeks to detect 0.1 mg/mL MG. The test results indicated 94% of its initial response, compared to 93% in Reference [17], which confirmed the satisfactory stability of the fabricated biosensor. Hence, the biosensor in this work has better selectivity, reproducibility and more desirable stability than [17].

### 4.3. Actual Samples Test

The mixture of MG samples and sand-stone water was applied to measure to validate the performance of the measurement system in sample testing. The different concentrations of MG in fishery water were measured by designed measurement system and the enzyme-linked immunosorbent assay (ELISA) method respectively, the comparison results are shown in Table 2. We found that, compared with the results of ELISA, the detection discrepancy of the measurement system was within 10%. It is indicated the reasonability of the measurement system for applications in complicated samples.

## 5. Conclusions

A novel multi-channel electrochemical malachite green measurement system has been successfully demonstrated and discussed, which is portable, low-cost and high-precision. The proposed measurement system provides a high-accuracy malachite green residue detection method to improve the measurement accuracy of electrochemical DNA biosensor. This system also presents the design of a hand-held electrochemical instrument to achieve an electrochemical chip with low cost, high accuracy and portability. The weak current detection method proposed in this system ensures the stability of scanning potential and eliminated the impact of parasitic parameters on the results, thereby improving the detection precision of the malachite green residues. The maximum measurement capability of multi-channel acquisition system achieves 16 samples in a batch. All of the measurement data can be transferred to cloud server wirelessly for data storage and processing. Extensive validation experiments indicate that this system performs well in the detection of malachite green residue in aquatic products transportation and the detection accuracy is less than 1 μg/L.

## Figures and Tables

**Figure 1 micromachines-09-00646-f001:**
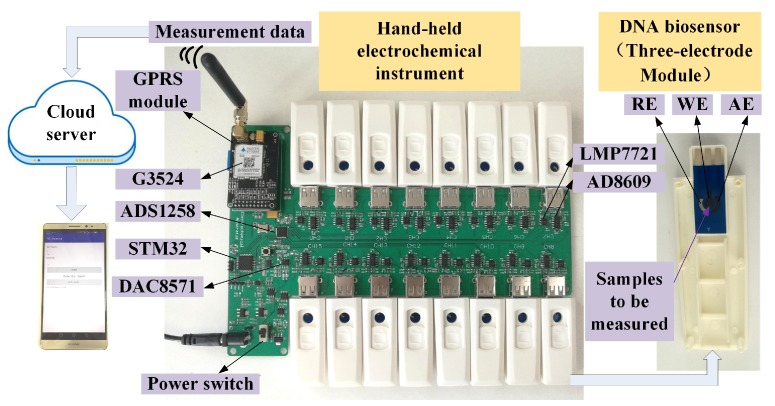
Image of the measurement system.

**Figure 2 micromachines-09-00646-f002:**
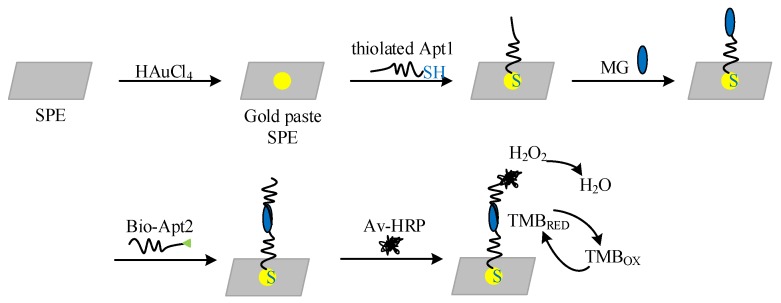
Schematic of the construction of the electrochemical DNA biosensor. SPE–screen printing electrode; MG–malachite green.

**Figure 3 micromachines-09-00646-f003:**
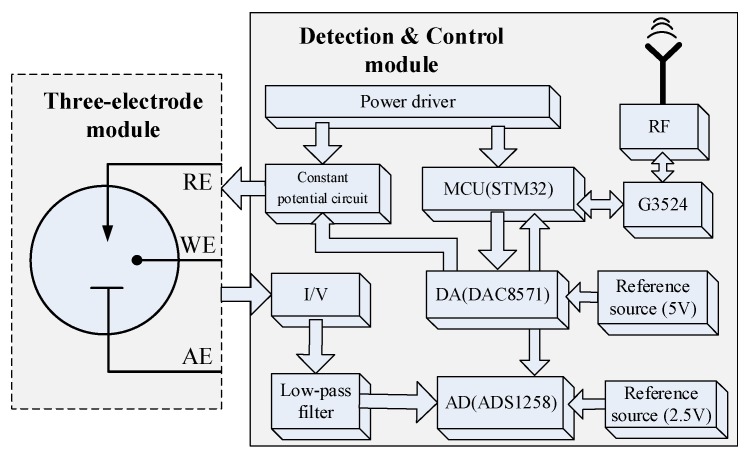
Measurement system structure schematic.

**Figure 4 micromachines-09-00646-f004:**
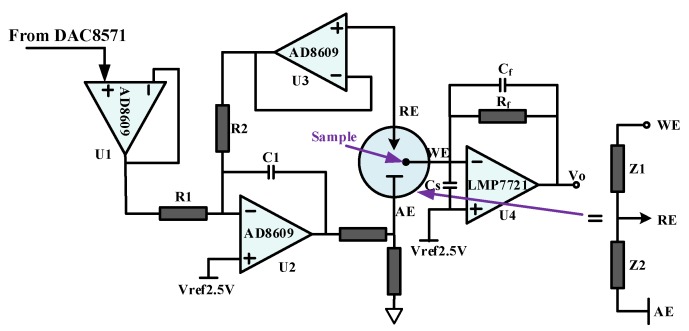
Circuit of control amplifier.

**Figure 5 micromachines-09-00646-f005:**
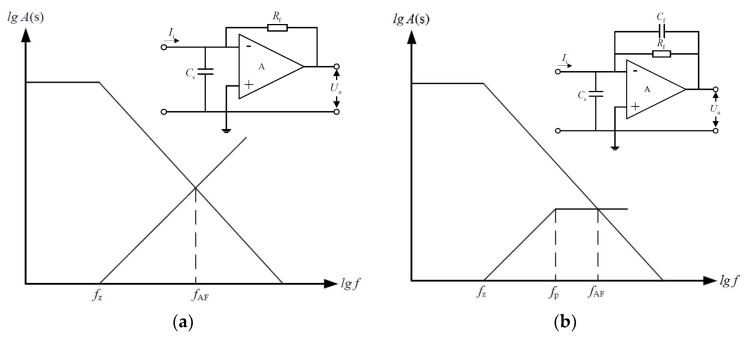
The amplitude frequency response of the open loop gain and the noise gain. (**a**) Amplifier without lead correction. (**b**) Amplifier with lead correction.

**Figure 6 micromachines-09-00646-f006:**
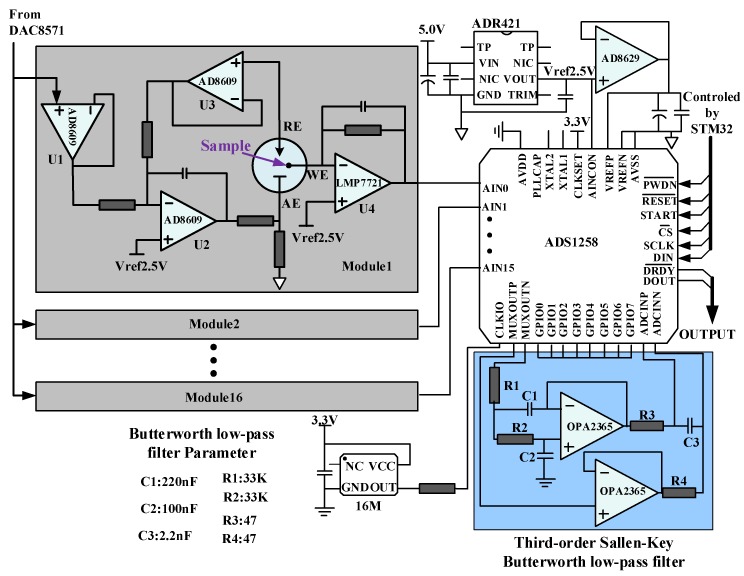
Multi-channel measurement circuit diagram.

**Figure 7 micromachines-09-00646-f007:**
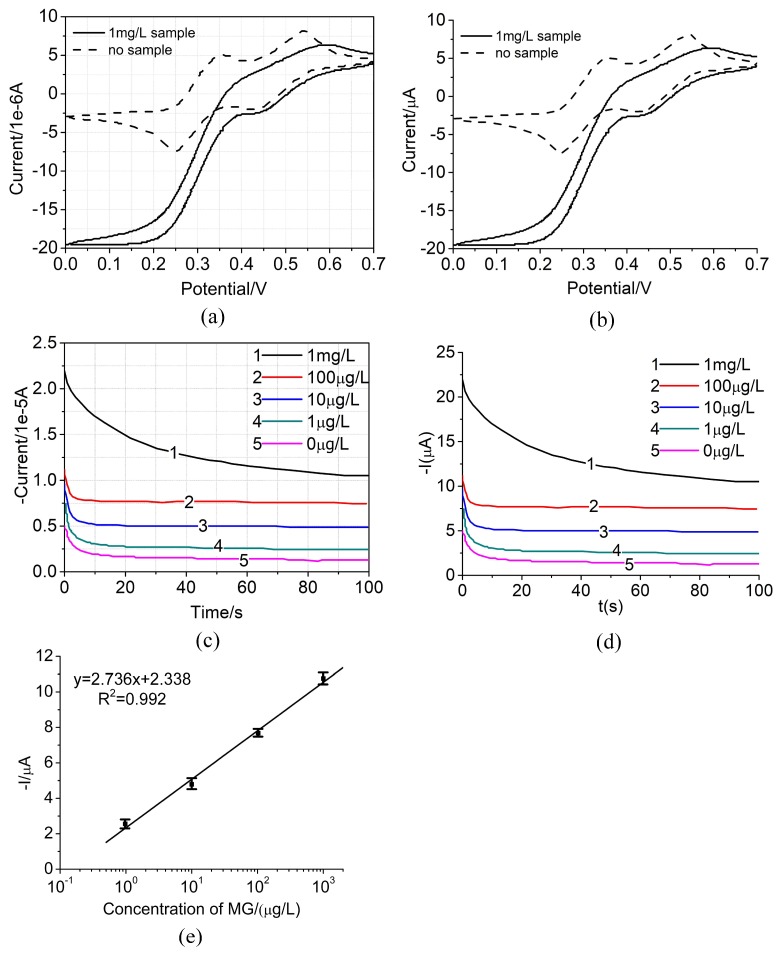
Detection performance of MG. (**a**) Cyclic voltammograms for no target (dashed line) and 1 mg/L target (solid line) obtained from commercial electrochemical instrument. (**b**) Cyclic voltammograms for no target (dashed line) and 1 mg/L target (solid line) obtained from the hand-held electrochemical instrument in this system. (**c**) Amperometric curves of samples with different concentrations (1 mg/L, 100 μg/L, 10 μg/L, 1 μg/L and 0 μg/L) in TMB substrate solution obtained from commercial electrochemical instrument. (**d**) Amperometric curves of samples with different concentrations (1 mg/L, 100 μg/L, 10 μg/L, 1 μg/L and 0 μg/L) in TMB substrate solution obtained from the hand-held electrochemical instrument in this system. (**e**) A calibration plot of the amperometric current and the log concentration of target. Data were collected from at least three independent experiments.

**Table 1 micromachines-09-00646-t001:** Constant current test results of 16 channels.

Measured Value	Mean of Channels/A	Standard Deviation of Channels/A
2 mA	0.0020062	10782 × 10^−6^
700 μA	0.0007025	9.203 × 10^−7^
200 μA	0.0002002	5.999 × 10^−7^
70 μA	6.995 × 10^−5^	1.542 × 10^−7^
20 μA	1.996 × 10^−5^	6.725 × 10^−8^
7 μA	6.986 × 10^−6^	1.087 × 10^−8^
2 μA	2 × 10^−6^	6.205 × 10^−9^
700 nA	7 × 10^−7^	1.426 × 10^−9^
200 nA	2.003 × 10^−7^	6.715 × 10^−10^
80 nA	8.037 × 10^−8^	7.706 × 10^−10^
12 nA	1.21 × 10^−8^	1.32 × 10^−10^
1 nA	1.055 × 10^−9^	1.02 × 10^−10^

**Table 2 micromachines-09-00646-t002:** Comparison of malachite green (MG) test results in experimental samples. ELISA–enzyme-linked immunosorbent assay.

Adding Standard (μg/mL)	Measured Quantity (μg/mL)	ELISA Measured Result (μg/mL)
$1.000\times 10^{-3}$	$(0.946\pm 0.045)\times 10^{-3}$	$1.029\times 10^{-3}$
$1.000\times 10^{-2}$	$(1.053\pm 0.049)\times 10^{-2}$	$0.992\times 10^{-2}$
$1.000\times 10^{-1}$	$\left(1.028\pm 0.052\right)\times 10^{-1}$	$1.034\times 10^{-1}$
1.000	0.985±0.050	1.033

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
