# Peer review of "High-Precision and Low-Cost Wireless 16-Channel Measurement System for Malachite Green Detection"

_micromachines, 2018, doi:10.3390/mi9120646_

Round 1

Reviewer 1 Report

The paper presents a wireless embedded system for the measurement of malachite green (MG) concentration. The authors discussed the development of the system and presented the results for MG detection both in laboratory artificial samples and real sand-stone water mixed with known concentrations of MG. I think the paper can be accepted if the authors comply with the following recommendations.

1)      Page 1. The sentence “Moreover, it is not convenient to employ these electrochemical instruments in motion detection” is not clear and must be reframed.

2)      Page 3. At the beginning of section 2.2 there is the sentence “… providing the necessary scanning voltage between WE and RE; collecting the induction current signals on the WE and converting them into the corresponding concentration value …”.  According to Figure 4 the current is collected at the WE but the voltage stimulus is applied between AE and RE (and not WE and RE).

3)      Page 4. At the beginning of section 3.1 the sentence “In the three electrode system, as the electrochemical reaction going on, the concentration of reactants on the electrode surface is decreasing and the concentration of products on the electrode surface is increasing, …” is not clear and must be discussed in more details. What the terms “reactants” and “products” refer to?

4)      Page 5. At lines 5 and 6 the feedback resistance is referred as R3 while in Figure 4 and equation 1 is referred as Rf. Please, use uniform notation.

5)      Page 5, line 6 from the end of the page: there is a reference to a critical frequency fAF. What is this frequency? What is its value?

6)      In section 4.2 CV measurements were carried out using a commercial electrochemical instrument, while amperometric measurements were carried out with both the commercial instrument and the proposed sensor system. However in section 2.2 it was stated that the proposed system can perform measurement using both CV and amperometry. Have the authors tried to carry out the CV measurements also with the proposed system? Is there any difference in the resulting CV curves?

Author Response

Dear Reviewer:

We are very grateful to you for giving us so many instructive comments, which are very useful for us to improve our manuscript. All changes made to the original manuscript are emphasized in "Track Changes". Based on these comments and suggestions, we have made detailed revisions on the original manuscript. You will find our point-by-point responses in the attached pdf file.

Reviewer 2 Report

See document attached

Author Response

(The authors gave the same response as above.)

Round 2

Reviewer 1 Report

The paper has been revised according to the reviewers comments, thus, in my opinion, is now suitable for publication. There are only few grammatical errors to correct.

-          In the title I think channal should be channel.

-          In the abstract (3rd line) there is the statement “ … DNA adapter …”. I think the correct statement is “ … DNA aptamer …”.

-          Page 8. There is the sentence “We found that the amperometric signal was logarithmically ….. as shown in Fig. 7 (d)”. In the new revision of the paper the figure is Fig. 7 (e).

Reviewer 2 Report

The authors have made a great job, have wisely responded all the referee's questions, and performed all suggested corrections. I believe that now the manuscript is suitable for the publication in Micromachines.